# A Comprehensive Review of Neuromuscular Manifestations of COVID-19 and Management of Pre-Existing Neuromuscular Disorders in Children

**DOI:** 10.3390/jcm11040934

**Published:** 2022-02-11

**Authors:** Daniel J. Goetschius, Yunsung Kim, Ashutosh Kumar, Dustin Paul, Sunil Naik

**Affiliations:** 1Penn State College of Medicine Medical Scientist Training Program, Hershey, PA 17033, USA; dgoetschius@pennstatehealth.psu.edu (D.J.G.); ykim3@pennstatehealth.psu.edu (Y.K.); 2Department of Pediatrics and Neurology, Penn State Health Children’s Hospital, Penn State College of Medicine, Hershey, PA 17033, USA; snaik2@pennstatehealth.psu.edu; 3Department of Neurology, University of Texas Rio Grande Valley, Edinburg, TX 78539, USA; djpaul262@gmail.com

**Keywords:** COVID-19, MIS-C, children, neuromuscular, pediatric, SARS-CoV-2, telemedicine, muscular dystrophy, spinal muscular atrophy, treatment

## Abstract

Since the emergence of SARS-CoV-2, several studies have been published describing neuromuscular manifestations of the disease, as well as management of pre-existing pediatric neuromuscular disorders during the COVID-19 pandemic. These disorders include muscular dystrophies, myasthenic syndromes, peripheral nerve disorders, and spinal muscular atrophy. Such patients are a vulnerable population due to frequent complications such as scoliosis, cardiomyopathy, and restrictive lung disease that put them at risk of severe complications of COVID-19. In this review, neuromuscular manifestations of COVID-19 in children and the management of pre-existing pediatric neuromuscular disorders during the COVID-19 pandemic are discussed. We also review strategies to alleviate pandemic-associated disruptions in clinical care and research, including the emerging role of telemedicine and telerehabilitation to address the continued special needs of these patients.

## 1. Introduction

As with most specialties, the ongoing COVID-19 pandemic has complicated the care of pediatric neuromuscular disorders (NMDs) in numerous ways. NMDs, especially where cardiorespiratory function is compromised, put patients at increased risk for severe disease. COVID-19 has in rare instances been associated with neuromuscular diseases, with rare reports of Guillain–Barre syndrome (GBS) or myasthenia gravis, more commonly in adults than children. As with other vaccines, rare cases of GBS have been reported after SARS-CoV-2 vaccination as well.

Additionally, the pandemic created disruptions in society that impact ongoing disease management. By June 2020, the Italian Association of Myology reported suspension of rehabilitation services and outpatient visits at 93% of centers, transitioning to telemedicine when possible, while maintaining urgent on-site visits [1].

Expert consensuses for conditions such as muscular dystrophies and spinal muscular atrophy were issued to help clinicians adapt to new situations [2,3]. For patients with physical therapy needs, lockdowns reduced mobility and increased sedentary behavior. Office visits were postponed or reduced. Reliance on telemedicine has increased, which was easier for follow-up visits than for new diagnoses. Centralized clinical trials carried increased risk, and efforts were made to modify and continue the trials via remote administration [4].

Early in the pandemic, an ad hoc group of specialists published a set of ethical principles regarding ongoing care of pediatric neuromuscular patients, especially in the setting of resource limitation secondary to the pandemic [5]. These guidelines emphasized transparency of triage algorithms and utilization of diverse perspectives in the development of policies for resource allocation. Eligibility for initiating or continuing care should be based on the known features of an individual patient’s conditions. To avoid perpetuating bias or inequality, institutions should avoid using pre-existing or future disability as a factor to prioritize resources. Pediatric patients with an anticipated prolonged recovery or those requiring resource-intensive treatment have similar chances of returning to their previous baseline, and should not be disadvantaged during triage. Individual neuromuscular conditions should be considered only if they result in a poor chance of survival to discharge. Lastly, triage officers should make ready use of clinicians and advanced practice providers with relevant expertise due to the broad range of pediatric neuromuscular conditions.

These patients have higher baseline medical needs. Anticipating continued supply chain disruptions, extra equipment, medicine, and supplies should be obtained when possible. Provision of backup caregivers should be prioritized [6].

In a survey of Dutch adults and parents of children with NMDs, most reported significant worsening of their biopsychosocial health and quality of life, despite only 14 adults and 1 child testing positive for COVID-19, all mild or asymptomatic [7]. Perceived worsening was attributed to disrupted access to care, social isolation, and mental health challenges such as anxiety and depression.

## 2. Neuromuscular Manifestations of COVID-19 in Children

Neurological manifestations of COVID-19 range from mild (headache, dizziness, anosmia, ageusia) to more severe, including Guillain–Barré syndrome (GBS), encephalopathy, encephalitis, acute disseminated encephalomyelitis, and stroke [8]. Specifically in children, a recent study that included 90 patients who tested positive for SARS-CoV-2 reported that 13 patients (14.4%) developed new onset neurological symptoms, and 4 patients had epilepsy exacerbations [9]. Neuromuscular manifestations due to COVID-19 are rare. One out of the 90 patients in the Sandoval et al. study presented with progressive ascending flaccid tetraparesis, areflexia, and cranial nerve palsies, and was diagnosed with an acute motor axonal neuropathy (AMAN) variant of GBS [9]. Four other case reports of GBS in pediatric patients with COVID-19 have been published in the literature. In three cases, patients presented with symptoms consistent with classic sensorimotor GBS, while the fourth case was an AMAN variant [10,11,12,13]. An additional 3 cases of GBS were reported in a series of 23 pediatric patients in Mexico presenting with neurologic manifestations [14]. New onset myasthenia gravis (MG) post-COVID-19 infection in a pediatric patient has also been reported. The patient developed acetylcholine receptor antibody positive ocular MG 72 h after resolution of multisystem inflammatory syndrome in children (MIS-C) secondary to COVID-19 [15]. Multiple cases of MG in adult patients after COVID-19 have been published [16,17]. Molecular mimicry between a SARS-CoV-2 peptide and the nicotinic acetylcholine receptor subunit alpha-2 has been suggested as a possible mechanism to directly link the development of MG secondary to SARS-CoV-2 infection; however, it remains unclear whether a direct causal relationship exists between the two conditions [18,19].

There have been numerous reports of COVID-19-associated rhabdomyolysis in pediatric patients [14,20,21,22,23,24,25,26,27,28,29] leading to acute kidney injury or acute renal failure [20,23,27,28,29]. In some cases, rhabdomyolysis was the presenting symptom [26,29]. Both myositis and rhabdomyolysis may occur as a part of MIS-C [22,30]. There have been several reports of acute benign childhood myositis in the pandemic period, attributed to influenza or COVID-19 [31]. Finally, there have been reports of increased incidence of juvenile dermatomyositis (JDM), which may be attributable to true dermatomyositis, prolonged post-viral myositis, or a dermatomyositis-like syndrome [32,33]. Uncertainty regarding the precise etiology of myositis is also noted within the adult literature [34,35,36].

## 3. NMDs as Risk Factor for Severe Disease in Children

While not universal, pediatric neuromuscular diseases often carry comorbidities that may put patients at risk of severe disease, including cardiomyopathy, restrictive lung disease, impaired cough function, aspiration risk, sleep-disordered breathing, and immunosuppressive drug use [6,37].

In the absence of COVID-specific risk stratification according to baseline lung function, Stratton et al. recommend using established guidance for respiratory tract infections (Table 1) [6]:

Real-world data have been divided. A study of 130 children admitted to Italian hospitals early in the pandemic showed that only a small percentage had severe or critical disease (8.5%; 6.9%). Significant comorbidities were present in 26% of children, most commonly respiratory, cardiac, or neuromuscular chronic diseases (12% overall). Despite the predominance of asymptomatic or mild-to-moderate disease, this study raised initial concern that comorbidities including NMDs may be a risk factor for severe disease in children [38].

Confounding evidence came from a small case series of 29 children with NMDs reported by the Neuromuscular Working Group of Spanish Pediatric Neurologic Society. This study did not show severe complications, with 89% categorized as mild/asymptomatic, and 10% as moderate. However, among these patients, relatively more severe COVID-19 was associated with spinal muscular atrophy (SMA) type 1 [39].

Likewise, a small case series of muscular dystrophy patients provides encouraging evidence that muscular dystrophy may not be as great a risk for severe COVID as feared. Of the 116 Duchenne/Becker muscular dystrophy patients at a tertiary neuromuscular center in Israel, COVID infection was reported in seven: six with DMD and one with advanced BMD. Only two were admitted for hospitalization, with one requiring nocturnal non-invasive ventilation. Despite comorbidities including severe restrictive lung disease, obesity, and chronic immunosuppression secondary to steroid use, all patients recovered [40].

A large, retrospective cohort study at 45 US children’s hospitals examined 4063 children and adolescents hospitalized with COVID-19. Neuromuscular conditions were associated with greater disease severity, as were cardiovascular, pulmonary conditions, and obesity/type 2 DM. Demographic factors included race (Black or other non-White) and age greater than 4 years. Likewise, neurologic disease was associated with an adjusted odds ratio of 3.2 for admission from the ED among 19,976 total encounters [41]. A case report from Saudi Arabia describes a 3-year-old female with spinal muscular atrophy type 1 causing generalized hypotonia and respiratory failure, who was infected with SARS-CoV-2 and presented with fever, GI symptoms, and pediatric acute respiratory distress syndrome. The patient was treated with oseltamivir, oral dexamethasone, IVIG, and tocilizumab, but progressed to MIS-C and died [42].

## 4. Telemedicine and Rehabilitation

### 4.1. Telemedicine

The pandemic resulted in an abrupt transition to telemedicine for many specialties. In the US, the March 2020 Coronavirus Aid, Relief, and Economic Security (CARES) Act relaxed restrictions on telemedicine, and allowed Medicare to reimburse telehealth at the same rate as office visits [43,44].

Adoption of telemedicine in pediatric neurology was rapid. In a June 2020 survey of 100 child neurologist attendings and nurse practitioners, almost 80% reported having not used telemedicine prior to the pandemic, yet over 90% transitioned within 3 weeks of office closure. Telemedicine was largely considered acceptable in the initial evaluation of some conditions, e.g., seizure, ADHD, headache (72–80%). However, it was universally seen as inappropriate in neuromuscular disease (0%). Telehealth follow-up of NMDs was more acceptable, at 36%. Patient age < 1 and 1–5 years were also seen as less favorable for initial diagnosis (43%; 82%) and follow-up (82%; 95%), compared with general acceptance in older age groups (99–100%) [45].

Telemedicine still has barriers for both providers and caregivers. On a technological level, barriers include dependable broadband Internet access, hardware requirements, technical support, and the broad diversity of platforms. On a practical level, telemedicine makes it difficult to conduct a comprehensive physical exam or to establish rapport with the patient and their family [44].

Despite these difficulties, modified versions of many neuromuscular evaluations for muscular dystrophies and SMA are possible. This includes the Performance of Revised Upper Limb Module, Northstar Ambulatory Assessment, CHOP INTEND, and Hammersmith Functional Motor Scale Expanded and Revised Hammersmith [6]. The use of modified exams must however be noted in the medical record, as they are not directly comparable with gold standard assessments.

As practices and resources continue to resume in-person activities, balance must be struck in order to maintain patient and provider safety. This includes through formal guidelines such as those released for neurophysiology labs in Canada, highlighting infection control procedures with individualized recommendations for EMG and nerve conduction studies [46].

### 4.2. Rehabilitation

A recent review laments the lack of high-powered randomized controlled trials on the efficacy of rehabilitation in neuromuscular diseases, but supports the likely safety of low-to-moderate intensity exercise programs and potential benefits to systemic health, including retention of function, and potential reduction in the progression of muscle weakness and aerobic capacity [47]. However, restrictions due to the pandemic have required modification of existing rehabilitation protocols as described in the disease-specific sections below.

Despite limited access to physical therapy, patients should continue existing regimens such as night splints, standers, and gait trainers, with daily stretching and positioning. Creative solutions may be required to maintain physical activity, such as bike rides, walks, or massages. Such solutions should be encouraged to prevent contractures or disuse atrophy [6].

To allow for the continuation of clinical trials during the pandemic, specialized neuromuscular physical therapists (PTs) developed guidelines for administration of remote physical therapy and evaluation. The guidelines address location of the patient, PT, clinical evaluator (CE), and involvement of a caregiver as assistant for the remote exam. Physical safety, data protection, and validity of clinical assessment were all addressed, with the authors reporting success in introducing remote visits to seven research trials over 20 trial sites [4].

### 4.3. Educational Needs

Many of these patients have disabilities covered under Section 5.4 of the Rehabilitation Act of 1973 and the Individuals with Disabilities Education Act (IDEA) of 1975, requiring special services or Individualized Educational Plans (IEPs). These challenges have only been aggravated by the pandemic, with transitions to and from remote education. Internet access and specialized computer interfaces are only two of the challenges that families have faced. Parents, educators, school districts, and therapists must continue to engage in frequent communication and problem-solving to address the multitude of challenges as they arrive. Priority should be placed on safe return to the classroom in the least restrictive environment possible [6].

## 5. Management of Pre-Existing NMDs in Children

### 5.1. Muscular Dystrophies and Myopathies

Duchenne and Becker muscular dystrophies are X-linked recessive NMDs that cause progressive weakness due to mutations in the gene that encodes for dystrophin, a support protein necessary for muscle integrity [48]. Many patients with Duchenne or Becker muscular dystrophies are on chronic corticosteroid treatment, potentially putting them at a higher risk for developing severe complications from COVID-19. Poor respiratory function, cardiomyopathy, scoliosis, and obesity, which are common comorbidities for this patient population, are also risk factors that can contribute to outcomes with COVID-19. Two studies have been published investigating muscular dystrophies as a potential risk factor for severe COVID-19 infection. Both studies reported mild to moderate symptoms for patients with minimal complications and full recovery [40,49]. Similar to other neuromuscular conditions, muscular dystrophies do not seem to predispose patients to severe manifestations of COVID-19.

In an expert panel consensus statement released in 2020, guidelines for care of patients with Duchenne and Becker muscular dystrophies were established [2]. Patients on chronic corticosteroids should continue their treatment and be educated on stress dosing of corticosteroids in the setting of acute sickness. Because of the potential interactions of angiotensin-converting-enzyme (ACE) with SARS-CoV-2, there were initially some concerns as to whether ACE inhibitors and related angiotensin receptor blockers (ARBs) would impact the course of COVID-19. However, multiple studies in adults have established that ACE inhibitors and ARBs do not have an impact on recovery from COVID-19 [50,51]. Patients should continue to take ACE inhibitors or ARBs as prescribed by their physician. Other modifications to the standard of care can be made to reduce the risk of exposure to COVID-19, such as at-home infusions for exon-skipping agents, individualization of monitoring of pulmonary, cardiac, and bone health, as well as telemedicine and telerehabilitation.

The main impact of the pandemic for patients with muscular dystrophies is their access to care. In a survey administered from May to June of 2020 in Japan, 30% of patients out of 542 responders reported delays in follow-up visits. Caregivers also reported that the closing of rehabilitation centers has led to a decrease in physical exercise and an increase in caregiver burden [52]. Because of decreased physical activity, one study found that patients with Duchenne muscular dystrophy had significantly decreased range of motion for ankle dorsiflexion, while body mass index and other motor functions were not significantly impacted [53]. To mitigate the decreased access to care during the pandemic, telerehabilitation programs have been developed. Sobieraksja-Rek et al. reported that online instructional videos were successful in teaching caregivers and patients how to perform exercises at home [54,55]. However, the authors noted overall low interest from the patients in continuing to do telerehab exercises, perhaps due to increased burden placed on their time. An at-home pulmonary function monitoring system was also trialed for a small group of patients with Duchenne muscular dystrophy, which showed adequate quality of pulmonary function testing. However, there was low compliance due to forgetting and lack of motivation [56]. For future implementations of telerehabilitation and at-home monitoring systems, more patient and caregiver education would be needed in order to establish an understanding of treatment and monitoring goals.

Similar strategies including telemedicine and telerehab to include modified home exercise programs can be adapted for patients with congenital myopathies.

### 5.2. Myasthenic Syndromes

As with any respiratory infection, COVID-19 presents a risk for profound worsening of respiratory muscle function in patients with myasthenia gravis or congenital myasthenic syndromes [6]. Exacerbations of existing myasthenia gravis have likewise been described in adult patients with COVID-19 [57,58,59]. Additionally, chronic immunosuppressive therapy may increase the risk of infection [37].

Little guidance exists specific to the management of myasthenic syndromes in pediatric patients. Early in the pandemic, the International MG/COVID-19 Working Group released guidelines largely recommending continuing management as usual, with individualized decisions between the patient and provider, prioritizing steps to avoid infection [60]. In early guidance from the French Rare Health Care for Neuromuscular Diseases Network (FILNEMUS), interruption of existing therapy was discouraged, including biologics such as rituximab, with the initiation of a new therapy to be decided on a case-by-case basis. IVIG should be administered at home. Hydroxychloroquine may cause an exacerbation and is not recommended [61].

Guidon et al. introduced the Myasthenia Gravis Core Exam (MG-CE), an adaptation of the physical exam to telemedicine. This can be administered via video visit in under 10 min, encompassing eight components: ptosis, diplopia, facial strength, bulbar strength, dysarthria, single breath count, arm strength, and sit to stand [62].

A simpler, 4-step evaluation was proposed by the Digital Technologies, Web and Social Media Study Group of the Italian Society of Neurology, featuring an online video protocol to monitor myasthenia patients. This consists of a counting aloud (CAT) and hoarseness test (HT), as well as a head up (HUT) and 3-oz swallowing test (ST) to be administered with the support of a trained caregiver [63].

Regarding vaccination, evidence for safety and efficacy in myasthenia gravis is currently limited. A single-center case series of 22 adult myasthenia patients supports the safety of vaccination in patients with Myasthenia Gravis Foundation of America (MGFA) classification I to II disease [64]. Limited evidence is available regarding the efficacy of vaccination in these patients [19], as so-called fragile patients with malignancy, neurological and immunological disorders were excluded from vaccination trials. Efficacy may be especially limited in myasthenic patients on immunosuppressive therapy, including targeted B-cell biologics. The VAX4FRAIL trial includes 100 myasthenia gravis patients in its neurological disorder wing and should provide more conclusive evidence for patients and providers [65].

### 5.3. Peripheral Nerve Disorders

There is no specific guidance on the management of peripheral nerve disorders in pediatric patients during the COVID-19 pandemic. While not specific to children, one study examined the effect of lockdowns on patients with Charcot–Marie–Tooth (CMT) mainly in Europe and North America [66]. Patients reported reduced mobility during lockdowns, as expressed in a significantly reduced number of walks per week. Patients also perceived a small but significant increase in leg and arm pain. The authors suggest the use of a telerehabilitation program and physician guidance to encourage continued physical activity. The same author provided a case report, likewise of an adult CMT patient, who was diagnosed with mild COVID-19 while recovering from tendon-transfer surgery [67]. There were no complications specific to the underlying NMDs. Hand rehabilitation was continued via telerehabilitation and ad hoc online evaluation.

### 5.4. Spinal Muscular Atrophy

SMA is a disease characterized by degeneration of motor neurons, leading to progressive muscle atrophy. One of the leading causes of morbidity and mortality in patients with SMA is respiratory failure, which results from the weakening of muscles involved in respiration. Furthermore, SMA patients are at a higher risk for respiratory tract infections in general [68], raising the concern for severe manifestation of COVID-19 in this population. Limited data on outcomes of COVID-19 in children with SMA exist in the literature. In a small observational study, 29 children with neuromuscular diseases and COVID-19 were assessed for symptom severity. A total of 90% of the patients had asymptomatic to mild COVID-19, while 10% developed moderate disease, described as frequent fevers and pneumonia. Out of the 29 patients, the most prevalent neuromuscular condition was spinal muscular atrophy, with six SMA type 1 patients and five SMA type 2. The authors of the study concluded that the protective role of young age seemed to outweigh the risk factors associated with having a neuromuscular condition [39]. Interestingly, a bioinformatics analysis investigating the possible interaction between the SMN gene and ACE2, a receptor involved in SARS-CoV-2 entry into the cell, suggests that SMN1 deficiency may modulate ACE2 levels and therefore have a direct impact on the pathogenesis of COVID-19 [69].

An expert panel consensus statement on the care of patients with SMA during the COVID-19 pandemic was published in April 2020. In this guideline, the authors emphasize the importance of following national and local guidelines as well as clarifying that the treatments for SMA should not be perceived as elective or nonurgent, and therefore not subject to delay [3]. Treatment options currently available for patients with SMA include nusinersen, an intrathecal inject of antisense oligonucleotide targeting SMN2 pre-mRNA, onasemnogene abeparvovec-xioi, a one-time intravenous infusion of AAV9 containing the SMN1 transgene, and risdiplam, a small molecular SMN2 splicing modifier administered orally. Nusinersen is given as four loading doses in the first 2 months followed by maintenance doses every 4 months. According to data from Biogen, a one-time delay of 1 month in maintenance dose resulted in a 10% decrease in CSF exposure to the drug. Higher nusinersen CSF exposure is correlated with greater efficacy [3]. If a dose is delayed, the current recommendation is to give the missed dose of nusinersen as soon as possible. In a small study conducted in Italy, 25 patients with SMA were characterized during the first lockdown period when 8 patients experienced a delay in nusinersen infusions with a median delay of 58 days (range 26–91). No correlation between delayed treatment and changes in functional scores was observed. However, 48% of parents perceived worsening in muscle strength which was attributed to suspension in physical therapy rather than delays in the infusion [70]. For patients receiving onasemnogene abeparvovec-xioi, who require oral corticosteroids for 2 months following the initial infusion, the concern lies with the impact of chronic corticosteroid use and potential for developing severe disease from SARS-CoV-2 infection. Experts agree that patients should continue to take oral corticosteroids; however, family members should be extra vigilant in social distancing and be educated on stress dosing of steroids in the setting of acute illness from COVID-19 [3,6]. Risdiplam was approved by the FDA in August 2020 for patients with SMA age 2 months and older [71]. Because it is given orally, delays in administration are not expected with this medication and may provide patients with an alternative therapeutic option especially in the setting of the pandemic.

## 6. Conclusions

During the ongoing COVID-19 pandemic, expert guidance should be followed for the management of patients with pre-existing SMA, muscular dystrophies, and myasthenia gravis. Where specific information is lacking, as with CMT, clinicians should maintain the existing standard of care and avoid delaying treatment, while taking steps to protect the safety of patients and staff. It is important to continue existing treatment and therapy regimens, with modifications to allow for telehealth as necessary.

These patients may present with high risk of severe COVID-19 due to the presence of cardiorespiratory comorbidities or immunosuppressive therapy, though the data is not fully conclusive. As a preventative measure, immunization should be considered. Immunization against COVID-19 is considered to be safe in NMDs, though evidence for its efficacy is currently lacking in patients with autoimmune disease or on immunosuppressive/immunomodulatory therapy. Unless another contraindication to vaccination exists, a history of GBS or autoimmune condition does not preclude vaccination against COVID-19 [72,73].

## Figures and Tables

**Table 1 jcm-11-00934-t001:** Risk stratification according to respiratory function.

Characteristics	Risk
FVC < 50% predicted Impaired cough function Established use of noninvasive ventilation or augmented airway clearance	Highest risk of acute respiratory failure, intubation, and death
FVC % predicted > 50% but <80% Moderately impaired cough function	Risk of severe respiratory disease
Episodic or waxing/waning disease (myasthenic syndromes)	Profound respiratory muscle weakening

## Data Availability

Not applicable.

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
