# Peer review of "A Comprehensive Review of Neuromuscular Manifestations of COVID-19 and Management of Pre-Existing Neuromuscular Disorders in Children"

_jcm, 2022, doi:10.3390/jcm11040934_

Round 1

Reviewer 1 Report

The article provides a very nice and comprehensive review of COVID-19-related NMD complications. The structure is clear. I find the article beneficial. My only suggestion is  to include the myopathy  chapter – the rhabdomyolysis and myositis cases – this part is missing.

Author Response

"The article provides a very nice and comprehensive review of COVID-19-related NMD complications. The structure is clear. I find the article beneficial. My only suggestion is  to include the myopathy  chapter – the rhabdomyolysis and myositis cases – this part is missing."

We appreciate the kind comments from Reviewer 1. In response to their suggestions, have added a section on rhabdomyolysis and myositis under "2. Neuromuscular manifestations of COVID-19 in children" (lines 90-98) with 17 additional references. This includes numerous case reports.

In addition, we have added three more reported cases of GBS in children with COVID-19 (lines 79-80) in 1 new reference.

Lastly, we have made several minor corrections in grammar and formatting to improve readability (lines 36, 41, 53, 85-87, 114, 161, 177, 179, 186, 191, 193, 251, 290, 291, 299).

Reviewer 2 Report

The manuscript submitted by Daniel J. Goetschius et al. represents a comprehensive review of the COVID-19 impact in children with neuromuscular disorders, including the manifestations of the COVID-19 infection but above all the consequences of the pandemics in non-infected children. The authors review and discuss different aspects (consequences of pandemics in relation to diagnosis, management, and specific treatments), the strategies to alleviate these problems (telemedicine and telerehabilitation as examples), and the reported experiences from different countries and in various diseases. From my point of view, the main contribution is the extensive review done by authors in comparison with previous reports, the inclusion of aspects that are not commonly discussed such as the consequences of delaying administration of specific medications or decreasing physical therapy, the patient perceptions, and the expert recommendations on relevant issues.

Author Response

"The manuscript submitted by Daniel J. Goetschius et al. represents a comprehensive review of the COVID-19 impact in children with neuromuscular disorders, including the manifestations of the COVID-19 infection but above all the consequences of the pandemics in non-infected children. The authors review and discuss different aspects (consequences of pandemics in relation to diagnosis, management, and specific treatments), the strategies to alleviate these problems (telemedicine and telerehabilitation as examples), and the reported experiences from different countries and in various diseases. From my point of view, the main contribution is the extensive review done by authors in comparison with previous reports, the inclusion of aspects that are not commonly discussed such as the consequences of delaying administration of specific medications or decreasing physical therapy, the patient perceptions, and the expert recommendations on relevant issues."

We appreciate the generous comments from Reviewer 2. We have made two minor additions to the manuscript, as well as several minor corrections in grammar and formatting to improve readability. We hope that these will make this review even more valuable.

We have added a section on rhabdomyolysis and myositis under "2. Neuromuscular manifestations of COVID-19 in children" (lines 90-98) with 17 additional references. This includes numerous case reports.

We have added three more reported cases of GBS in children with COVID-19 (lines 79-80) in 1 new reference.

Minor corrections in grammar and formatting have been made (lines 36, 41, 53, 85-87, 114, 161, 177, 179, 186, 191, 193, 251, 290, 291, 299).